# The Association between Masticatory Muscles Activation and Foot Pressure Distribution in Older Female Adults: A Cross-Sectional Study

**DOI:** 10.3390/ijerph20065137

**Published:** 2023-03-14

**Authors:** Giuseppe Messina, Alessandra Amato, Federica Rizzo, Ligia Juliana Dominguez, Angelo Iovane, Mario Barbagallo, Patrizia Proia

**Affiliations:** 1Department of Psychology, Educational Science and Human Movement, University of Palermo, 90128 Palermo, Italy; 2Sport and Exercise Sciences Research Unit, Department of Psychology, Educational Science and Human Movement, University of Palermo, 90144 Palermo, Italy; 3Posturalab Research Institute, 90144 Palermo, Italy; 4Geriatric Unit, Department of Medicine, University of Palermo, 90100 Palermo, Italy; 5School of Medicine and Surgery, University of Enna “Kore”, 94100 Enna, Italy

**Keywords:** exercise therapy, postural balance, dental occlusion, muscles, surface electromyography, aging, neurology

## Abstract

The association between craniofacial muscles and postural control is well-known because of numerous anatomical connections. However, there are a few conflicting studies that correlated the activity of the masticatory muscles with the distribution of body weight pressure on the feet, which can strongly influence balance. Therefore, the purpose of our study was to evaluate the association between the masseter and temporalis muscle activity and foot pressure distribution. Fifty-two women were recruited, and baropodometric and EMG analyses of the masseter and temporalis baseline activities were analyzed. An ipsilateral association was found between the right temporal muscle activity and the right rearfoot load (r = 0.29, *p* < 0.05) and right forefoot load (r = −0.29; *p* < 0.05), as well as the percentage of muscle activation of the right masseter muscles with the percentage of pressure on the right forefoot (r = 0.31, *p* < 0.05) and right rearfoot (r = −0.31, *p* < 0.05). While further studies are needed, an ipsilateral association was found between masticatory muscles and foot pressure distribution.

## 1. Introduction

### 1.1. The Postural Control

Postural control, defined as the muscular activity that controls anteroposterior and later-lateral oscillations and the distribution of the pressure load of body weight at the feet, allows us to maintain stability in both static and dynamic positions; this is especially important in older adults, where the loss of static and dynamic balance, and thus, falling is common [1]. Several strategies are used to achieve good balance, especially in these populations, such as training the muscles directly involved in postural control [2]. Under normal conditions, the foot pressure distribution should be 50% between the right and left foot and about 60% and 40%, respectively, for the rearfoot and forefoot [3,4]. The alteration of connective tissues, which characterizes the aging process and is possibly accelerated by the presence of other diseases, such as neurodegenerative diseases with onset in old age, alters the distribution of plantar pressure and this correlates with static balance and gait monitoring test results [5,6]. In turn, the plantar pressure distribution seems to be influenced by other neuromotor and biomechanical factors, such as occlusion.

### 1.2. The Relationship between Postural and Masticatory Muscles

More specifically, the position of the bone component of the temporomandibular joint (TMJ) resulting from masticatory muscle activity influences the activity of both proximal postural muscles, such as those of the neck region, and distal ones, such as those of the lower limbs, which allow body weight to be distributed to the feet. Bergamini et al. analyzed the rest activities of paired sternocleidomastoids, the erector spinae at the L4 level, and soleus muscles in subjects with malocclusions and found a beneficial effect of balancing the occlusion on the postural muscles. Therefore, dental occlusion muscles affect the activity of other muscles [7]. The authors hypothesized that this association is because of complex neuro-mediated reflexes with supreme coordination controllers, which give a possible positive influence of occlusion muscle manipulation on the physiology of the postural system [7].

### 1.3. Tools for the Masticatory Muscles Activity Analysis

However, studies gave very conflicting results, especially on what tools to use to assess these associations. Dysfunctions and temporomandibular muscle activity are evaluated with several techniques, such as devices that measure extension or contraction force or pressure (e.g., Myoscanner 160-B) [8]. However, previous researchers investigated the influence of occlusion on body posture by analyzing surface electromyography (sEMG) of the masticatory muscles [9,10,11,12] and this tool is considered the gold standard for analyzing the tonicity, activity, and symmetry of these muscles, in particular, the main muscles deputed to the elevation of the jaw, such as masseter and temporalis [13] considered in our study. Baldini et al., who in their study evaluated the use of a force platform to assess the association between dental occlusion and posture, stated that the force platform does not constitute the most ideal method for the analysis of the association between dental occlusion and posture because of the lack of sensitivity of the method, especially in healthy subjects, and that the sway area seems to be the only sensitive parameter for evaluating the effect of occlusion on body posture [14], confirming the results found from Perinetti et al. [15] in the previous year [15]. However, a few studies explored the impact of occlusion by considering the neuromuscular component of the variables of foot pressure distribution evaluated with the force platform. Therefore, the main objective of our work is to analyze whether there is an association between EMG parameters of the temporal and masseter muscles and posturographic parameters from body weight pressure distribution analysis with a baropodometric platform.

## 2. Methods

### 2.1. Recruitment Process

For our cross-sectional study, we randomly recruited subjects by telephone from the list of patients (80 subjects) who had been in care at the geriatric ambulatory clinic of the Department of Internal Medicine and Geriatrics of the University Hospital of Palermo (Palermo, Italy) in the last five years, following an alphabetical order as in the list. The list included patients only over the age of sixty and we decided to call only those who were under 80 as per one of the inclusion criteria. We asked whether they met the inclusion criteria we had previously established; if they met these criteria, we then asked whether they were willing to undergo postural and electrical activity assessments of the masticatory muscles at the postural laboratory of the geriatric clinic of the University Hospital of Palermo (Italy). The recruitment process was carried out in December 2019, and the evaluation section was scheduled from the end of December 2019 to the end of February 2020.

### 2.2. Inclusion and Exclusion Criteria

The following inclusion criteria were applied:Aged between 60 and 80 years old;No head or face trauma in the 6 months before evaluation;Had not undergone any invasive interventions in the mouth in the 6 months before evaluation (prostheses, implants, extractions, devitalization);No falls or trauma history in the previous 6 months;Had habitual occlusion with fair balance at contact points in both subjects with prostheses (fixed, mobile) and without prostheses.

The following exclusion criteria were applied:Not meeting the inclusion criteria listed above;Not being able to travel to the clinic on the scheduled day of evaluations;Not being able to perform all the tests administered independently.

All participants were administered a surface EMG assessment of the masseter and temporal muscles during a maximum natural intercuspation through a wireless device capable of measuring the balancing of dental occlusion (Teethan, Garbagnate Milanese, Milano, Italy) and a baropodometric evaluation to assess the plantar pressure distribution in an orthostatic position using a baropodometric platform (FreeStep, Sensor Medica, Guidonia Montecelio, Roma, Italy). Each subject underwent the two assessments on the same day. All analyses were done by the same operator at the geriatric ambulatory clinic of the University Hospital of Palermo (Italy). Informed consent was obtained from all subjects involved in the study.

### 2.3. EMG Assessment

Before the performance, the subject was asked to remove all metal objects they were wearing to avoid interference with the electromyograph. Then the test was explained and shown. The “Teethan” surface electrodes were applied at the origin and insertion of the masticatory muscles, specifically in the muscle belly of the anterior temporalis and those of the masseters. The subject was seated in a chair with the sacrum and shoulder blades well placed on the backrest, hands resting on the thighs, and feet firmly on the floor. The performance of the examination involved two phases, each lasting 5 s: the first phase was the calibration, in which two cotton rolls were placed between the upper and lower arches at the height of the first molar. The second, performed in natural intercuspation, allowed for the parameters of the occlusal board to be measured. At the end of each 5-second trial, the “Teethan” processes the recorded muscle activity values, ensuring objectivity and repeatability of the measurement. Left and right activation (%) and the percent overlap coefficient (POC) (%) were taken into consideration for each muscle from the EMG analysis.

### 2.4. Baropodometric Analysis

The analyses of foot pressure distribution (Baropodometric Analysis) were performed with the FreeMed posturography system (produced by Sensor Medica, Guidonia Montecelio, Roma, Italy), which is a pressor platform with conductive rubber resistive-type sensors. In 40 cm × 40 cm configurations, it interfaces with the computer through a USB port connection; it develops a sampling frequency of up to 400 Hz in real time. All participants stood barefoot on the platform for 5 s with the most comfortable position of the feet looking straight ahead with their arms along their sides. The following parameters were evaluated:The feet surface areas (cm^2^);Forefoot and hindfoot load (%);Left and right foot load (%);Feet angle (°);Pressure points (gr/cm^2^).

### 2.5. Statistical Analysis

Statistical analysis was performed using the software “SPSS” version 23.0 (IBM SPSS Statistics 23).

The EMG and baropodometric analysis variables were expressed as mean ± standard deviation. The association between variables was analyzed using Pearson’s correlation coefficient. The a priori sample size was calculated using “G*Power 3.1” (α = 0.05, β = 0.8).

## 3. Results

From 80 recruited, 8 patients were excluded because they did not meet the inclusion criteria, 11 were excluded because they were not available to perform the assessments at the clinic on the days of evaluations, and 9 subjects were excluded because they did not complete the baropodometric analysis without support; therefore, 52 women (aged between 60 and 80, height 156.65 ± 7.2 (cm), weight 62.27 ± 8.3 (kg)) were included in the study (Figure 1).

Table 1 summarizes the results of the postural analysis and electrical activity of the masseter and temporalis muscles. The table shows the descriptive statistical analysis for each variable.

A positive correlation was found (r = 0.29; *p* < 0.05; 95% CI, 0.04 to 0.52) between the percentage of muscle activation of the right temporal muscle and the percentage of pressure on the right rearfoot (Figure 2A); consequently, the same but negative correlation (r = −0.29; *p* < 0.05; 95% CI, −0.52 to −0.04) was found between the percentage of muscle activation of the right temporal muscle and the percentage of pressure on the right forefoot (Figure 2B). Regarding the percentage of muscle activation of the right masseter muscles, a positive correlation was found with the percentage of pressure on the right forefoot (r = 0.31; *p* < 0.05; 95% CI, 0.04 to 0.54) (Figure 2C), along with the consequent negative correlation with the percentage of pressure on the right rearfoot (r = −0.31; *p* < 0.05; 95% CI, −0.54 to −0.04) (Figure 2D).

No significant correlation was found when crossing all the other baropodometric analysis and EMG analysis variables (Table 2).

## 4. Discussion

### 4.1. Anatomical Association between Trigeminal Nerve Activity and Vestibular Nuclei

The masseter and temporalis muscles are antigravity muscles rich in neuromuscular spindles that allow them to perform a precise function: to maintain a basic muscle tone to counteract the force of gravity to keep the jaw elevated. They are part of the stomatognathic system and innervated by the trigeminal nerve. Their neuromuscular spindles are afferent to the trigeminal motor nucleus, which regulate the tone of the masseter and temporalis muscles by receiving feedback from the neuromuscular spindles [16]. These structures act in harmony to perform various functional tasks, such as swallowing and mastication, predominantly via the activation of these two muscles [17]. Thus, while trigeminal innervation to the masseter and temporalis muscles is responsible for jaw posture, the vestibular nuclei are responsible for the modulation of the activity of the muscles deputed to balance control. The literature strongly demonstrates an anatomical connection between these two structures [18,19,20]. The trigeminal motor nucleus projects to the ipsilateral cerebellum and contralateral thalamus [19,21,22], which suggests that portions of the trigeminal system strongly influence the coordination of posture. Shi et al. [23] reinforced the hypothesis of neuroanatomical interconnection between the trigeminal mechanism of somatic movement and balance, reiterating the existence of an ascending pathway that links proprioceptive activity from the periodontal region to the cerebellum. In fact, Shi et al. [23] found that modifying occlusion in the analyzed sample caused altered function of the cerebellum, impacting the balance of body posture; specifically, they showed that altered occlusion influences neuronal excitation of the cerebellum through the mesencephalic nucleus trigeminal–cerebellum circuit [23]. These strong anatomical interrelations of the trigeminal mesencephalic nucleus–cerebellum circuit highlight that a pathological condition in one area can likely alter the function of the other and cause associated symptoms, such as body posture imbalance.

### 4.2. Temporomandibular Disorder and Change in Foot Pressure Distribution

Postural adjustments are the result of a complex system of mechanisms controlled by multisensory inputs embedded in the central nervous system (CNS). Furthermore, feed-forward mechanisms, which also involve neuromuscular spindles, are deputed to maintain postural control. These adjustments are evoked by different types of afferent inputs, which are also modulated by the stomatognathic system (SS) [24]. Indeed, several studies showed a link between TMJ disorders and stomatognathic system dysfunctions with foot disorders and a change in foot pressure distribution [25,26]. Therefore, Cuccia et al. [24] tested different TMJ positions, and consequently different levels of masticatory muscle contraction, and found an association with a change in the pressure loads and surface areas for both feet. In particular, the condition of voluntary tooth clenching determined a load reduction and an increase in both feet’s surface contact. Instead, when subjects had the cotton rolls in their mouth, there was a change in the load distribution between the forefoot and rearfoot [1]. The extensive afferent and efferent innervations of the stomatognathic system are reflected in the extensive representation of the orofacial district in the motor and sensory areas of the cerebral cortex [27]. Consequently, alterations in the cranio-cervical-mandibular region, such as the TMD pain system, affect human posture, resulting in impaired balance control. A postural disorder can have implications for occlusion due to the interactions of anatomical pathways. Therefore, our study aimed to find the association between an excessive basal activation of the masseter and temporalis muscles and an altered foot pressure distribution.

Our results suggest an ipsilateral association both for temporalis and masseter muscles with the feet’ pressure load distribution. The right temporalis muscle activation was positively correlated with the right rearfoot load (*p* < 0.05) and negatively correlated with the right forefoot (*p* < 0.05); conversely, the right masseter muscles activation was positively correlated with the right forefoot load (*p* < 0.05) and negatively correlated with right rearfoot load (*p* < 0.05). The temporalis muscle is a posterior muscle, while the masseter, on the other hand, is an anterior muscle; therefore, the association between temporalis activation and load on the rearfoot could be a consequence of the shortening of the entire posterior muscle chain, and vice versa of the anterior muscle chain for the masseter, given the anatomical connection. Additionally, Myers et al. and Stecco et al. demonstrated that the fascial system is connected such that changes occur due to these multiple connections, for example, an anterior cruciate ligament injury can generate changes in the masseter, and vice versa [28,29]. This reflects on changes in postural posture and foot pressure distribution and could justify the ipsilateral relationship between masticatory muscles activation and foot pressure load detected in our study. Indeed, in 2008, Rothbart et al. theorized an ascending foot–cranial model to explain the association between foot position and vertical facial dimension, confirming the relationship between the ipsilateral nerve pathways [30]. In addition, this can consequently lead to overloading of the respective joints, such as the TMJ, parafunctional habit, or temporomandibular muscle hypertonia, which is associated with the onset of joint degeneration [31], as well as hip [32] and knee [33] joint alteration. Therefore, intervening by restoring a correct basal tone of the masticatory muscles could potentially improve postural control and also be useful in older adults who already have age-related alterations, such as weakening of the connective tissue with atrophy of the plantar fat pad, which may lead to the loss of the natural buffering properties of the forefoot and local pressure peaks, with several consequent problems [34].

This study had some limitations: this was a cross-sectional study, and thus, we cannot draw any conclusions regarding causality; we did not classify subjects using intercuspation class, which may have influenced the resting position that was required for the EMG analysis, which could be slightly different from one subject to another with possibly different degrees of masticatory muscle tension; and only women were selected for our study to avoid the possible influence of the complex sex factors

## 5. Conclusions

Our findings suggest an ipsilateral association between plantar pressure load distribution and the activation of the masticatory muscles in older adult women. Moreover, this association also seemed to depend on the anterior or posterior placement of the muscles. Although further studies are needed, the future implication of our study could be to use hypnotism in an intervention program to improve occlusion and evaluate the effect on the distribution of foot pressure distribution.

## Figures and Tables

**Figure 1 ijerph-20-05137-f001:**
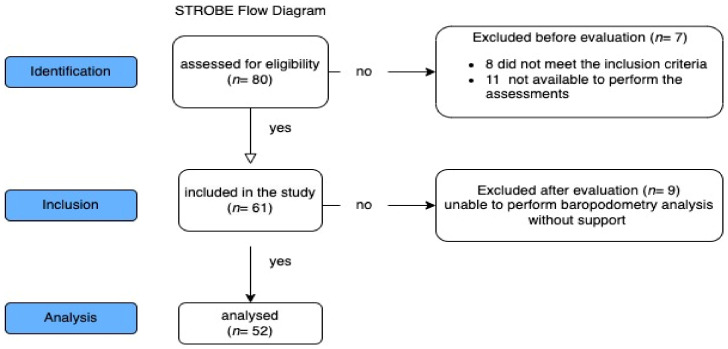
The STROBE flowchart describing the recruitment process.

**Figure 2 ijerph-20-05137-f002:**
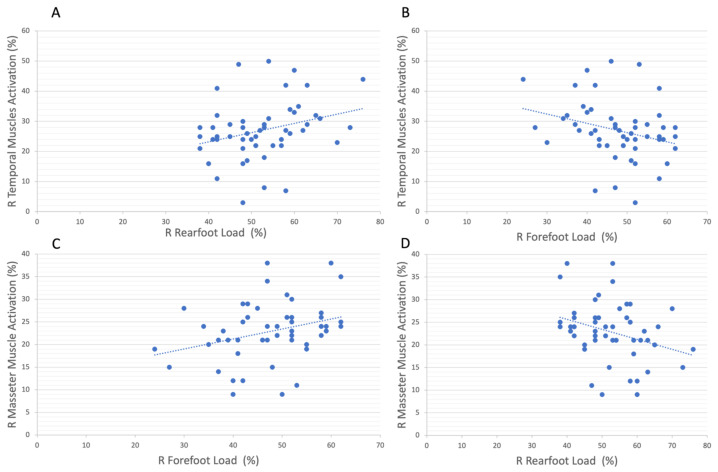
The scatterplot shows the correlations between the (**A**) right temporal muscles activation (%) and the right rearfoot load (%) (correlation: r = 0.29, conf.int = 0.95), (**B**) right temporal muscle activation (%) and right forefoot load (%) (correlation: r = 0.29, conf.int = 0.95), (**C**) right masseter activation and right forefoot load (%) (correlation r = 0.31, conf.int= 0.95), and (**D**) right masseter activation and right rearfoot load (%) (correlation r = 0.31, conf.int = 0.95). R: right, L: left.

**Table 1 ijerph-20-05137-t001:** Postural and EMG variables. Variables are expressed as mean and standard deviation. R: right side of muscle; L: left side of muscle; POC: percent overlap coefficient (it identifies the overall activity of both masticatory muscles).

	**Female *n* = 52**	
	M	SD
POC temporal muscles (%)	80.88	8.75
POC Masseter Muscle (%)	81.93	8.29
R Temporal Muscles Activation (%)	26.96	9.77
L Temporal Muscles Activation (%)	26.12	7.42
R Masseter Muscle Activation (%)	22.90	6.44
L Masseter Muscle Activation (%)	23.25	8.67
L Surface (cm^2^)	104.77	68.33
R Surface (cm^2^)	99.94	65.61
L Forefoot Surface (cm^2^)	57.71	38.62
R Forefoot Surface (cm^2^)	54.62	36.02
L Backfoot Surface (cm^2^)	46.94	30.30
R Backfoot Surface (cm^2^)	45.33	29.97
L Foot Load (%)	51.42	5.31
R Foot Load (%)	48.58	5.31
L Forefoot Load (%)	46.46	9.95
R Forefoot Load (%)	47.62	9.12
L Rearfoot Load (%)	53.54	9.95
R Rearfoot Load (%)	52.38	9.12
L Max Pressure Point (gr/cm^2^)	709.38	160.39
R Max Pressure Point (gr/cm^2^)	692.23	150.33
L Average Pressure Point (gr/cm^2^)	337.08	79.62
R Average Pressure Point (gr/cm^2^)	332.19	68.58
L Feet Angle (°)	6.85	3.82
R Feet Angle (°)	7.08	3.56
L Feet Axis (°)	6.69	3.96
R Feet Axis (°)	8.63	5.67

**Table 2 ijerph-20-05137-t002:** Correlation matrix between EMG and baropodometry variables. *: significant correlation *p* < 0.05.

		**Postural Variables**															
**EMG Variables**		**R Surface (CM2)**	**L Surface (cm^2^)**	**R Forefoot Surface (cm^2^)**	**R Backfoot Surface (cm^2^)**	**L Backfoot Surface (cm^2^)**	**R Rearfoot Load (%)**	**R Foot Load (%)**	**R Forefoot Load (%)**	**L Forefoot Load (%)**	**L Rearfoot Load (%)**	**R Max Pressure Point (gr/cm^2^)**	**L Max Pressure Point (gr/cm^2^)**	**L Feet Angle (°)**	**R Feet Angle (°)**	**L Foot Load (%)**	**L Forefoot Surface (cm^2^)**
Poc Temporal Muscles (%)	Pearson Correlation	−0.026	−0.047	−0.042	−0.004	−0.053	0.188	0.231	−0.188	0.008	−0.008	0.048	−0.037	−0.040	−0.001	−0.231	−0.043
	Sig.	0.857	0.743	0.769	0.978	0.709	0.182	0.100	0.182	0.957	0.957	0.737	0.794	0.778	0.995	0.100	0.763
Poc Masseter Muscle (%)	Pearson Correlation	−0.006	−0.038	−0.010	0.001	−0.043	−0.056	0.157	0.056	0.123	−0.123	0.043	0.019	0.129	−0.111	−0.157	−0.039
	Sig.	0.967	0.789	0.941	0.994	0.761	0.694	0.266	0.694	0.384	0.384	0.760	0.892	0.361	0.433	0.266	0.784
R Temporal Muscles Activation (%)	Pearson Correlation	−0.043	0.005	−0.065	−0.016	0.008	0.287 *	−0.168	−0.287 *	−0.026	0.026	0.032	0.074	−0.067	0.010	0.168	0.003
	Sig.	0.761	0.969	0.649	0.912	0.955	0.039	0.235	0.039	0.852	0.852	0.821	0.602	0.637	0.943	0.235	0.983
L Masseter Muscle Activation (%)	Pearson Correlation	−0.014	−0.014	−0.009	−0.022	0.001	−0.119	−0.045	0.119	−0.112	0.112	0.044	0.059	−0.023	−0.050	0.045	−0.025
	Sig.	0.923	0.922	0.951	0.876	0.992	0.400	0.750	0.400	0.431	0.431	0.757	0.678	0.874	0.724	0.750	0.861
L Temporal Muscles Activation (%)	Pearson Correlation	0.167	0.160	0.158	0.178	0.151	0.092	0.103	−0.092	0.166	−0.166	−0.057	−0.117	0.080	0.184	−0.103	0.166
	Sig.	0.237	0.256	0.265	0.208	0.286	0.518	0.466	0.518	0.240	0.240	0.686	0.408	0.571	0.192	0.466	0.239
R Masseter Muscle Activation (%)	Pearson Correlation	−0.147	−0.204	−0.121	− 0.178	−0.200	−0.313*	0.203	0.313 *	−0.051	0.051	0.026	0.011	0.042	−0.086	−0.203	−0.204
	Sig.	0.297	0.148	0.394	0.208	0.154	0.024	0.149	0.024	0.718	0.718	0.857	0.938	0.765	0.544	0.149	0.146

## Data Availability

Data is unavailable due to the privacy restrictions policies of the University Hospital of Palermo.

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
