# Peer review of "The Association between Masticatory Muscles Activation and Foot Pressure Distribution in Older Female Adults: A Cross-Sectional Study"

_ijerph, 2023, doi:10.3390/ijerph20065137_

Round 1
Reviewer 1 Report
Thank you for the opportunity to review this manuscript. I have read it with great interest.
The authors performed a study on the associations between masticatory muscles activation and 2 weight pressure foot distribution in older adults.
I have reviewed the manuscript and would like to make some major suggestions on how to improve the quality of reporting.
Title & Abstract:
- Please revise according to the comments below.
Introduction/Background:
- This section is very hard to read without any paragraph. Please revise.
- I recommend to use the term “association” instead of “correlation” throughout thew hole manuscript.
- Please avoid the stigmata term “elderly” and use “older adults” etc.
Methods:
- Please report according to STROBE guideline and provide a completed STROBE checklist within the revision.
- Please indicate the study design.
- Please report if the study was registered a priori.
- Please report inclusion and exclusion criteria explicitly.
- Please report the recruitment process in a much more detailed and transparent way.
- “We enrolled 52 women”. This is a result.
- “aged between 60 and 80 years old” Is this a result or inclusion criterion?
- Please provide a sample size calculation/justification.
Results:
- Abbreviations in TAB 1 needs explanation.
- Please add 95% for all correlation coefficients.
- Please include a (STROBE) flow chart.
Discussion:
- Needs to be revised according to the above stated recommendations/comments
- Please include paragraphs for better reading.
- Please include a Limitations section.
Further general comments:
- Although I myself am not a native English speaker, I feel that the manuscript would benefit from a professional/native English editing.
- Please indicate if all participants provided written informed consent.
Reviewer 2 Report
In line 69-76
Therefore, the main objective of our work is to analyze if there are correlations between EMG parameters of temporal and masseter muscles and posturographic parameters in order to hypothesize that an intervention that improves occlusion may have an effect on the distribution of body weight at the breech level and thus in preventing the risk of falls in elderly.
The presented part of the hypothesis
that an intervention that improves occlusion may have an effect on the distribution of body weight at the breech level and thus in preventing the risk of falls in elderly.
It is only a presumption, it was not confirmed in the realized study
In line 204-208
Conclusion
Although further studies are needed, our findings suggest an ipsilateral relationship between plantar pressure distribution and the activation of the masticatory muscles in elderly women without fall history and these may have a future implication for reducing the age-related risk of falls.
Presented conclusion
these may have a future implication for reducing the age-related risk of falls.
It is only a presumption, it was not confirmed in the realized study
I rekomande add the application conclusion of the implemented study.
Reviewer 3 Report
Interesting study that addresses new approaches and new relationships that influence postural control.
There are some issues that need to be reviewed or explained:
P-2 L 78
We enrolled 52 women aged between 60 and 80 –
How was the sample selected? Sequential? convenience?
P-2 L 79
Patients were recruited from the geriatric ambulatory clinic. Is it a general practice or is it specialized in problems with craniofacial muscles or postural control?
P-2 L 81
To be included in the study they did not have to have:
Language difficult to understand.
It is suggested:
The following exclusion criteria were applied:
P-2 L85
had any falls or trauma of any kind in the previous 6 months
Already included in the inclusion criteria “without fall history from” L-78
P-3
Table 1 (L 135)
It would be better if there was a separation between the EMG assessment variables and the Baropodometric analysis variables.
P-4 L 137 a 139
Multiple variables are shown, but only 1 is shown.
L 137-139:"A positive correlation was found (r=0.29, p p<0.05) between the percentage of muscle activation of the right temporal muscle (26.96  9.77) and the percentage of pressure on the right rearfoot ( 52.38 9.12)."
If the remaining variables were not compared or analyzed, why are they presented?
This analysis should be more extensive using the other variables collected.
P-5 L- 159-161
Indeed, anatomical connections between trigeminus nuclei and numerous other nuclei (e.g. vestibular, cervical spinal cord, oculomotor…)
The meaning of this sentence must be clarified. It seems incomplete.
P-6 L204-208
Conclusion
“Although further studies are needed, our findings suggest an ipsilateral relationship between plantar pressure distribution and the activation of the masticatory muscles in elderly women without fall history and these may have a future implication for reducing the age-related risk of falls.”
Preventing falls is important, but it is necessary to identify the risk, and identify which factors are contributing to an increased risk.
It is true that the risk of falling increases with age, but not only that, and we do not know the mean age of the sample either.
This conclusion seems premature to me. After all, the evaluated patients had no history of falls. Because the "ipsilateral relationship between plantar pressure distribution and the activation of the masticatory muscles" has implications for the prevention of falls. It should be better explained.
Authors should also not forget that causality is not studied, but only a relationship that is bi-directional.
General comments:
The title, abstract, introduction and the conclusion refers to the risk of falling and preventing falls, however, the risk of falling was not assessed. At least it is not mentioned in the article.
It was also left to explain how the relationship between the variables will have an effect on the prevention of falls.
